# An Overview of Substrate Copper Trace Crack Through Experiments, Characterization, and Numerical Simulations

**DOI:** 10.3390/mi16040428

**Published:** 2025-04-02

**Authors:** Wei Yu, Faxing Che, Vance Liu, Raymond Chen, Sam Ireland, Yeow Chon Ong, Hong Wan Ng, Gokul Kumar

**Affiliations:** 1Micron Semiconductor Asia Operations Pte. Ltd., 990, Bendemeer Road, Singapore 339942, Singapore; 2Micron Memory Taiwan Co., Ltd., Taichung City 42152, Taiwan; 3Micron Technology, Inc., 8000 South Federal Way, Boise, ID 83716, USA

**Keywords:** copper foil, copper trace crack, finite element analysis (FEA), material characterization, reliability testing

## Abstract

The high input/output demands of memory packages require precise trace width and spacing, posing challenges for contemporary package design. Substrate copper trace cracks are a major reliability issue during temperature cycling tests (TCTs). This study offers a detailed analysis of copper trace crack mechanisms through experimental observations, material characterization, and numerical simulations. Common failure modes of trace cracks are identified from experimental data, pinpointing initiation sites and propagation paths. Young’s modulus of copper foil samples is assessed using four testing methods, revealing consistent trends across samples from different substrate suppliers. Sample A with higher E/H values tested via nanoindentation correlated with lower failure rates in the experiment. Stress–strain testing on copper foil was successfully performed at the lower TCT temperature limit of −65 °C, providing vital input for finite element (FE) models. The simulations show strong alignment with trace crack locations under different failure modes. The impact of copper trace width and material properties is illustrated in numerical models by comparing variations in plastic strain responses, which show differences of up to 40% and 30%, respectively. The simulation design of the experiments (DOE) indicates that high-strength solder resist (SR) can significantly enhance temperature cycling performance by reducing SR and copper trace stress and strain by up to 75%. The accumulation of plastic strain in copper traces is predicted to increase up to four times when SR breaks at the crack location, underscoring the importance of SR in copper trace reliability.

## 1. Introduction

Copper (Cu) trace reliability is crucial in electronic packages, especially with the trend of minimizing trace line width and spacing while maximizing silicon content for better memory storage. The cracking of copper traces is a major concern during temperature cycling tests (TCTs), leading to product malfunction. The characterization of copper foil thermomechanical behavior is necessary to understand the failure mechanisms of such phenomena. Conducting actual experiments and detection of substrate copper trace cracks is often time- and resource-consuming; therefore, finite element analysis (FEA) simulations are useful for correlating thermomechanical stress with crack incidents. The consideration of plastic behavior in substrate material is necessary to determine the amount of plastic strain accumulated during TCTs and assess the risk level of the package design. This study aims to provide an overview of the package substrate copper trace crack mechanism and prevention through package-level reliability experiments, Cu foil material characterization, and numerical simulations on package reliability assessment under TCTs.

Copper is extensively utilized in the semiconductor industry due to its outstanding properties, including electrical and thermal conductivity, mechanical performance, and manufacturability. In semiconductor devices and advanced packaging applications, copper is primarily produced in the form of foil or thin films. This paper focuses on copper trace reliability in package substrates under temperature cycling testing conditions. Researchers have explored the intrinsic properties of copper foil, with process parameters significantly affecting its microstructure, which, in turn, influences properties like coefficient of thermal expansion (CTE), mechanical properties, and fatigue behavior [1]. Li et al. studied how copper ion concentration impacts microstructure and mechanical properties [2]. Wunderle et al. reviewed Cu thin film sample preparation and testing methods, noting that yield stress and stiffness increase as film thickness decreases due to changes in microstructure and grain size [3,4]. Fatigue tests on Cu films reveal that fatigue behavior is strongly dependent on film thickness, grain size, and interface characteristics [5]. Beck et al. fitted crack initiation data with the Coffin–Manson model, considering trace width and thickness [6]. Zhang et al. compared tensile and low-cycle fatigue properties of variously manufactured copper films, highlighting significant differences in Young’s modulus, yield stress, elongation, and crack resistance [7]. High-temperature conditions bring copper creep into focus for deformation and failure analysis [8,9].

Experiments show various trace crack failure modes from both the die and ball sides [10]. Excessive intermetallic compound (IMC) growth can lead to copper pad cracks in the secondary layer [11,12]. Finite element (FE) simulation studies on substrate copper trace cracks, using global–local modeling, offer high efficiency and accuracy. Simulations help with failure analysis, material selection, and design optimization, showing a strong correlation between predicted stress and actual failure locations [13]. Some studies highlight the complexity of fiberglass and localized stress-induced trace cracks. Abbas et al. [14] used FEA modeling and the Taguchi method to show that micro-vias in resin-rich areas are more susceptible to failure during reflow due to high stress from CTE mismatch compared to those in fiber-rich areas.

Substrate copper trace crack is a complex reliability involving multiple materials, not only copper but also other substrate components like SR and prepreg (PPG). Therefore, thorough considerations are essential to address this intricate issue. Beginning with experimental observations, we proceed to investigate material characterization and conduct analyses through numerical simulations. This study provides a systematic approach to understanding substrate Cu trace crack issues.

## 2. Experimental Observations

TCTs are an essential reliability requirement, particularly for automotive applications. Substrate trace crack failure rate and failure mode can vary significantly based on board-level settings and system operating conditions. The presence of underfill can impact the stress distribution within substrate layers, altering the failure mode from the ball side to the die side [15]. Substrate supplier-dependent performance differences are observed even with identical package designs due to variations in trace width and thickness, as well as the intrinsic material properties of copper foil, which can influence fracture toughness and trace crack performance in the temperature cycling test [14].

In the experiment, a combination of electrical failure analysis (EFA) and physical failure analysis (PFA) is essential for detecting copper trace crack issues. Initially, EFA identifies the malfunction signal channel, followed by PFA to determine the exact location of the trace crack within a complex multi-layer substrate. Experimental observations of various trace crack events have shown that these cracks commonly occur around silicon die stacks, either at the die edge or under die shadows, similar to previous research on trace cracks [16,17,18]. Many of these instances are related to the high silicon content of the package, which leads to greater CTE mismatch with the substrate and PCB.

Substrates are complex laminated structures composed of polymeric and metal materials. Solder resist (SR) and prepreg (PPG) resin material typically crack earlier than copper layer due to their lower ductility and strength at low temperatures. Cracking occurs when stress exceeds the threshold of the polymeric material. Once SR and PPG are damaged, the exposed copper trace becomes more susceptible to cracking due to the lack of protection. Crack initiation in substrates can occur from either the die side or the ball side. Several typical failure modes are summarized in Table 1. Failure mode A shows that the crack originates from the die side and near the base die edge at the top SR layer. Failure mode B originates from the bottom side of the substrate, which can either initiate from the bottom SR layer, leading to cracks at the secondary trace (B1), or penetrate into the PPG, causing cracks at the inner layer copper trace (B2). Currently, EFA is only able to detect signal abnormalities caused by copper trace cracks. However, it cannot accurately determine the cycle of cracking for SR and PPG. This presents challenges in developing fatigue life assessments using numerical simulations. Failure mode C is related to IMC growth, reducing Cu thickness, which is associated with bismuth-type hard solder and organic solderability preservative (OSP) surface finish.

## 3. Copper Foil Characterization

### 3.1. Sample Preparation

Copper foil with a thickness of 15 µm was provided by suppliers A and B. The sample preparation process is shown in Figure 1. Initially, electroless deposited (ED) copper was applied to the carrier, followed by an electrolytic-plated Cu layer. An annealing process was conducted prior to acid rinse and final detachment of the Cu foil samples. The Cu foil sheet was cut into different sizes for different types of tests. The thickness of ED copper, electrolytic-plated Cu, and annealing conditions are specified in Table 2 for samples A and B. Different annealing temperatures potentially cause different microstructure and material properties. Compared to conventional bulk copper and single-processed copper foil from previous studies, this sample more accurately represents the actual conditions of substrate copper, resulting in a better correlation with the experimental outcomes.

Electron backscatter diffraction (EBSD) demonstrates distinct profiles between two copper layers (ED copper and electrolytic-plated copper) before being subjected to TCTs, as illustrated in the cross-sectional view in Figure 2. Underneath, ED copper shows a smaller grain size than electrolytic-plated copper. The grain size and distribution of plated copper are influenced by the type of solution, current density, and annealing conditions used by each substrate supplier. As plated copper is in contact with the solder resist material, its material properties are more significant and important in trace reliability performance compared to the underlying ED copper.

A change in grain size is observed under repetitive temperature loading at the crack location near the base die edge. Figure 3 shows the EBSD results after thousands of TCT cycles for failure mode A. The grain boundaries at the crack location near the base die edge are mis-orientated compared to the area underneath the die shadow, showing a sign of high plastic strain. This is due to localized hardening under cyclic bending moments during TCTs. The enlargement of grain size leads to a shorter propagation length. Based on the grain orientation, the crack appears to initiate from the top surface of the electrolytic-plated copper facing SR.

### 3.2. Testing Methodology

Most copper trace crack failures are believed to occur at the low-temperature limit of temperature cycling, within the range of −40 °C to −65 °C, when the polymeric material becomes brittle and the thermomechanical stress due to material CTE mismatch is exceptionally high. Copper traces are exposed once the polymeric layers covering the top of Cu traces lose their integrity. Previous research primarily focused on stress–strain behavior at room temperature and high temperatures, with limited work on the low-temperature characterization of copper films. This study successfully obtained test data at −65 °C, covering the lowest TCT temperature limit at condition C (TC-C). Four different test methods were used to characterize these samples in terms of Young’s modulus, stress–strain behavior, and E/H (ratio of modulus to hardness). The information about the samples and test conditions are listed in Table 3. Tensile mode tests follow the testing standard of ASTM E345-16 [19]. Tensile tests using a dynamic mechanical analyzer (DMA) were performed to obtain stress–strain curves, serving as a critical input to FE simulation for describing the plastic behavior of substrate copper. Young’s modulus was also tested in the temperature sweep mode using the tensile mode of the DMA. A loading–unloading tensile test was conducted using a universal tester. Nanoindentation tests were performed on the plated copper layer to compare fracture toughness-related properties, such as the hardness (H), modulus (E), and ratio of E/H.

### 3.3. Results and Discussion

#### 3.3.1. DMA Stress–Strain Tensile Test

Stress–strain curves of copper foil were successfully obtained at 3 temperatures. Figure 4 shows typical temperature-dependent stress–strain curves from tensile tests for both samples. Young’s modulus was calculated based on the slope of the curve from a 0% to 0.1% strain range. The copper foil sample from supplier B shows higher ultimate tensile strength (UTS) and yield strength. The averaged Young’s modulus of samples A and B are listed in Table 4. Sample A shows a lower modulus at each temperature than sample B due to its coarser microstructure, as shown in Figure 5. Different microstructures result from varying manufacturing process parameters and annealing treatments by different vendors.

The higher yield strength of sample B corresponds to a smaller grain size, as shown in Figure 5. The relation between the metal’s yield stress and grain size can be described mathematically by the Hall–Petch [20] equation below:(1)$$\sigma_{y}=\sigma_{0}+\frac{k_{y}}{\sqrt{d}}$$
where $\sigma_{0}$ is the original yield stress, *d* is the average grain diameter, and $k_{y}$ is a constant.

#### 3.3.2. DMA Temperature Sweep Test

The temperature sweep mode test using a DMA was conducted for samples A and B using the conditions listed in Table 3. The temperature-dependent modulus can be easily obtained with this method over a wide temperature range using a single test piece. For comparison with the tensile testing results, Table 5 lists the storage modulus at the same temperatures. Similarly, sample B shows a higher storage modulus than sample A at the same temperature. The absolute value of the modulus obtained from the temperature sweep test is higher than that from stress–strain tensile test due to the high frequency used in the temperature sweep test.

#### 3.3.3. Loading—Unloading Test

Substrate copper trace crack in TCTs is a fatigue-induced failure under cyclic conditions; a cyclic loading–unloading test with strain range or stress range control is a more suitable method for investigating Cu trace fatigue performance. Such a testing method is also widely used to obtain the elastic modulus of materials for small-size samples such as thin films [1,21]. A 5-step loading–unloading tensile test, as shown in Figure 6, was applied for copper foil in this study. In the unloading stage, material elastic deformation recovers. Young’s modulus is calculated based on the average slope of the stress and strain in the unloading stage of curves 2 and 4. Young’s modulus at 3 temperatures is listed in Table 6 for samples A and B. The modulus from loading–unloading tests shows lower temperature dependence, maybe due to handling issues with longer Cu foil samples at high and low temperatures. Nevertheless, sample B exhibits a higher modulus than sample A, which shows a trend consistent with previous tensile tests using a DMA.

#### 3.3.4. Nanoindentation Test

Nanoindentation tests [22] on the plated copper layer were conducted at room temperature using the continuous stiffness measurement (CSM) method. Hardness (H) and modulus (E) values were calculated based on an indentation depth range of 200 nm to 300 nm. Although sample B exhibits a slightly higher modulus than sample A, it also demonstrates greater hardness, resulting in a lower E/H value, as shown in Table 7. The E/H ratio was used to evaluate the material’s fracture toughness, as shown in Equation (2). The lower E/H of sample B leads to lower fracture toughness, probably due to the lower annealing temperature used in the manufacturing process. This corresponds with the experimental findings that sample B has a higher failure rate under TCTs than sample A.(2)$$K_{C}=\beta {\left(\frac{E}{H}\right)}^{\frac{1}{2}}\frac{P}{c^{\frac{3}{2}}}$$
where $K_{C}$ is the fracture toughness, β is the empirical constant, E is the elastic modulus, H is the hardness, P is the peak load, and *c* is the crack length.

#### 3.3.5. Modulus Comparison

Based on the average Young’s modulus obtained using the four test methods at 30 °C, sample B consistently exhibits a higher modulus compared to sample A, as shown in Figure 7. This indicates intrinsic material property differences from varying suppliers, which may lead to supplier-dependent performance under identical package designs. The modulus value obtained in this study mostly falls in the 70~110 GPa range at room temperature. It shows a good agreement with previous studies for copper foil material [1,2,3,4,5,6,7,8,9,21,23].

## 4. Numerical Simulation

### 4.1. Modeling Methodology and Results

Numerical simulation is a valuable tool for visualizing stress distribution and understanding the complex failure mechanisms of trace cracks. The global–local modeling technique [10,13] is used in this study since it is useful for focusing on localized trace crack failures from package stress analysis. TCT condition C (TC-C) was implemented in the following FEA simulation with a temperature range of −65 °C to 150 °C. The transfer of displacement and boundary conditions from the global model to the local model ensures that the local model reflects the overall structural behavior while providing detailed insights into critical areas of trace crack analysis. There is a good correlation between the predicted stress–strain distribution and the actual trace crack locations (pointed by red arrows) observed in the experiments, as presented in Table 8, under various failure modes from the die side and the ball side. The simulation results show multiple hot spots in red, which represent high-stress/strain locations that are potential failure locations. However, in the actual failure analysis, the failure site may only be one site, such as mode B. The reason is that when the crack happens, energy is released, the stress is concentrated in the crack tips, and the stress in other locations is reduced. For mode C, cracks can initiate at high-stress locations, propagate to large cracks, and connect to cause failure. The simulation results from the local model predict potential failure locations that are consistent with the experimental findings. However, this simulation method and results for stress are static analysis results, which can be interpreted as a potential crack failure initiation point. In the real package reliability test, the final failure analysis includes information about crack initiation and propagation. Such a gap between the simulation and experimental results can be integrated by static stress simulation and fracture mechanics simulation to predict failure mode and reliability life, which will be one of the important future works.

### 4.2. Effect of Trace Dimension

Supplier B, with a thinner finished trace width of 20 µm, is observed to have a Mode A trace crack after TC-C 1000 cycles in the primary layer, as shown in Figure 8a, and the crack location is found at the base die edge, whereas supplier A, with a wider finished trace width of 30 µm, does not have such failure. To further understand the effect of trace width on trace reliability performance, FEA modeling and simulation were conducted by considering 4 different Cu trace width designs ranging from 15 µm to 30 µm. Figure 8b presents four simulation models and plastic strain results. The maximum plastic strain increases with a reduction in Cu trace width. The high strain location in model 2 has a good agreement with the failure location of the Cu trace, as shown in Figure 8a.

The maximum equivalent plastic strain values are normalized relative to a baseline model with a 25 µm width (model 3), as shown in Table 9. The simulation predicts that a wider copper trace of 30 µm (model 4) exhibits 13% lower plastic strain, whereas a narrower copper trace of 15 µm (model 1) leads to 27% higher strain compared to the baseline, adding up to a 40% difference in plastic strain response within the possible manufacturing tolerance range. The increased plastic strain in the narrower trace suggests a greater tendency for crack initiation. Additionally, a narrower trace has a shorter crack propagation path once a crack initiates. Based on simulation results, controlling trace width tolerance is crucial for improving trace crack performance under TCT conditions.

### 4.3. Effect of Copper Plasticity

To demonstrate TCT performance dependency on suppliers, Table 10 compares the effect of copper plasticity input along with the trace width effect. Model 4 and model 2a exhibit a 22% difference in volume-averaged copper plastic strain based on actual trace width and thickness measurement data from two substrate suppliers. Using the same geometry of copper trace in model 2a and model 2b, inputs of material properties from samples A and B are applied using a multilinear kinetic hardening command to iterate plastic strain accumulation based on test data from Figure 4. A 30% difference is predicted between these two inputs, correlating with experimental observations that supplier B has a higher failure rate after TCTs. Therefore, the characterization of copper foil is important for generating accurate simulation responses with accurate material properties.

### 4.4. Effect of SR Type

Due to the sequential failure nature of substrate trace cracks, solder resist plays a crucial role in protecting the underlying copper traces. As shown in Figure 9, the stress–strain curves obtained at −65 °C exhibit a very limited plastic region because the polymeric material becomes brittle at low temperatures. Therefore, UTS at −65 °C determines the ability to protect the underlying copper traces. Compared to SR type 1, the high-strength SR type 2 shows an advantage in the low-temperature range in terms of its UTS, which is effective in preventing copper trace cracks during prolonged cycles of TCTs.

The simulation evaluated Design A and Design B using different SR types, with the relative stress–strain response of SR and Cu trace compared in Table 11. The results indicate that Design B with SR type 2 shows significant stress–strain reduction of up to 75% under both failure modes A and B2 models because of its low CTE characteristics (close to copper). The experimental results confirm the effectiveness of SR type 2 in prolonging TCT performance across both failure modes. This suggests that for package designs with identical structures and bills of materials, altering the solder resist type can be a critical factor in enhancing TCT performance. High-strength SR types are particularly desirable in fine-pitch designs to meet stringent TCT requirements.

### 4.5. Effect of SR Breaks

The effect of SR breaks has been underestimated in previous modeling works. Traditionally, the focus has been solely on Cu plastic strain. Coffin–Manson’s law [24,25] is often used to relate Cu strain plastic behavior to fatigue life. SR is typically assumed to be unbroken in simulation models. However, recent experimental studies have shown that the presence of SR significantly influences the fatigue life of the underlying Cu traces [26]. This study proposed to use progressive modeling to demonstrate the significance of the SR layer using type 1 material, which consists of two progressive models:Model 1: This model represents the scenario when the SR layer is intact, with all elements being properly connected to each other.Model 2: This model simulates the damaged SR layer, where the copper trace is exposed and unprotected at the base die edge. This is achieved using the element kill function to deactivate a row of SR elements at the base die edge, on top of and surrounding the Cu traces, like a comb structure in Figure 10.

The plastic strain response of the Cu trace was extracted and compared between two models. The simulation indicates that changes in plastic strain accumulation are more significant with a higher SR thickness of 20 µm. The amount of plastic accumulation increases substantially (by up to four times) when SR breaks, which is indicated by a steeper slope of the curve. The simulation results using progressive modeling agree with the experimental study conducted by Chen [26], showing that a damaged window on top of copper traces leads to notably poorer fatigue life under bending.

The total plastic strain accumulation on the Cu trace significantly depends on the time of SR breaks. Figure 11 demonstrates three different scenarios with varying relative timing of SR breaks. This suggests that the fatigue life of the Cu trace is influenced by both copper plastic strain accumulation and the durability of SR. If a high-strength SR type could be used, the expected Cu plastic strain accumulation will be significantly retarded, as validated in the experimental results shown in Table 11. Therefore, advanced monitoring techniques are required to determine the actual time of SR failure under TCT conditions.

The development of fatigue life equations must consider the sequential failure of materials along the crack initiation and propagation path. The conventional Cu fatigue life model based on Coffin–Manson’s law is limited in describing the dynamic processes of substrate trace crack mechanisms. The simulations primarily predict the likelihood of crack initiation; however, crack propagation is closely related to the path length, which depends on factors such as copper thickness, microstructure, grain size, and surface roughness. Both propagation and initiation should be analyzed using Darveaux’s method [27].

## 5. Conclusions

This paper offers a comprehensive analysis of the substrate copper trace crack mechanism, incorporating experimental, material characterization, and numerical simulation perspectives. Typical failure modes on both the die side and the ball side are summarized based on extensive experimental observations, revealing common crack locations around and beneath the silicon die stack. Cracks typically initiate from the SR layer before propagating to the copper trace layers.

This study characterizes 15 µm thick copper foil samples from different suppliers, including ED copper and electrolytic-plated copper. Young’s moduli obtained using four different testing methods show a consistent trend between samples A and B, with sample B exhibiting a higher modulus compared to sample A due to its microstructure with a fine grain size. The modulus values and stress–strain curves obtained at −65 °C serve as critical inputs for subsequent simulation models. The E/H ratio derived from nanoindentation demonstrates a good agreement with the experimental results.

Numerical simulation using a global–local modeling approach is efficient and accurate in visualizing stress distribution in critical trace crack regions. There is a strong correlation between the experimental and simulation results in terms of high-stress/strain location and crack failure location. This study examines the effect of trace width changing from 15 µm to 30 µm and intrinsic copper plasticity, leading to 40% and 30% variation in plastic strain from simulation output, respectively. The importance of the SR type is discussed from both simulation and experimental perspectives on various failure modes. The effect of SR breakdown is demonstrated using a progressive modeling approach. Significant changes (up to four times) in copper plastic strain are predicted before and after SR breakdown, underscoring the necessity of developing a fatigue model based on sequential failure mechanisms.

## 6. Future Work

Future research should address the limitations of the current experiments, particularly in the characterization of solder resist fatigue. Understanding sequential failure mechanisms using effective monitoring methods from experiments is crucial. Additionally, the continuous development of copper trace fatigue life with a concrete database is necessary. Trace width tolerance control and process-dependent fracture toughness enhancement need further collaboration with substrate vendors. Local design features and routing techniques should also be considered to determine critical regions for trace cracks, ensuring a comprehensive risk assessment from both simulation and design perspectives. Integrated simulation of crack initiation and propagation using the fracture mechanics concept is also worthy of exploration to predict copper trace crack fatigue life.

## Figures and Tables

**Figure 1 micromachines-16-00428-f001:**
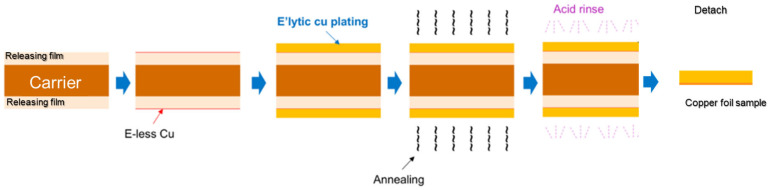
Schematic of Cu foil sample preparation process.

**Figure 2 micromachines-16-00428-f002:**
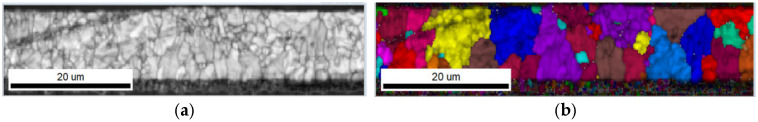
Microstructure of substrate copper foil at time zero. (**a**) Image quality map. (**b**) Grain size.

**Figure 3 micromachines-16-00428-f003:**
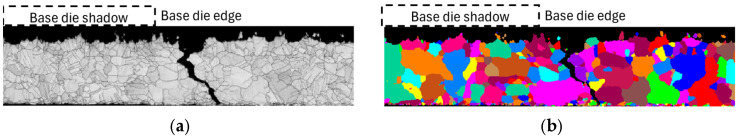
Microstructure of substrate copper foil after TCTs. (**a**) Image quality map. (**b**) Grain size.

**Figure 4 micromachines-16-00428-f004:**
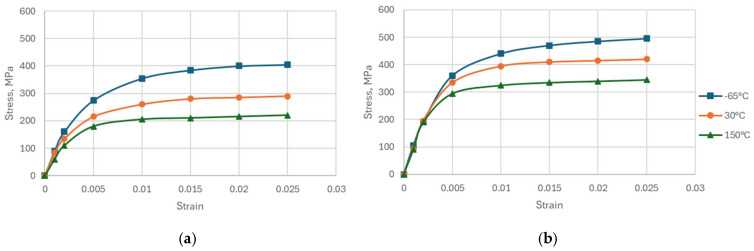
Stress–strain curves of copper foil. (**a**) Sample A and (**b**) sample B.

**Figure 5 micromachines-16-00428-f005:**
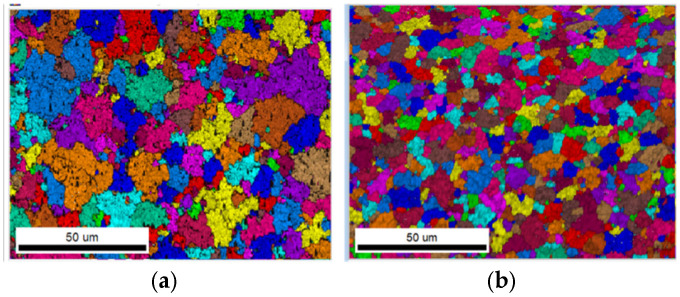
Grain size comparison from the top view of the plated Cu layer. (**a**) Sample A and (**b**) sample B.

**Figure 6 micromachines-16-00428-f006:**
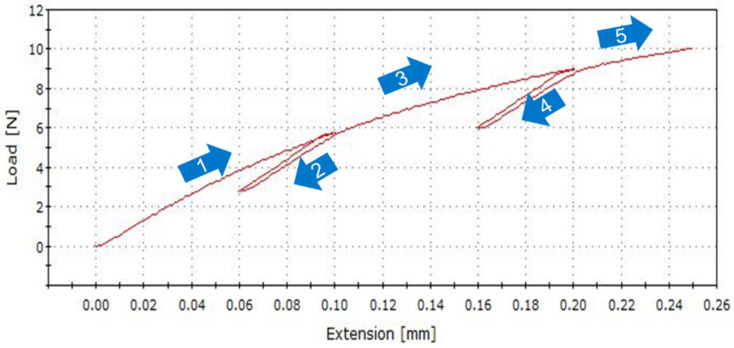
Typical testing curves of copper foil in loading–unloading tensile test showing loading steps of 1, 3, and 5 and unloading steps of 2 and 4.

**Figure 7 micromachines-16-00428-f007:**
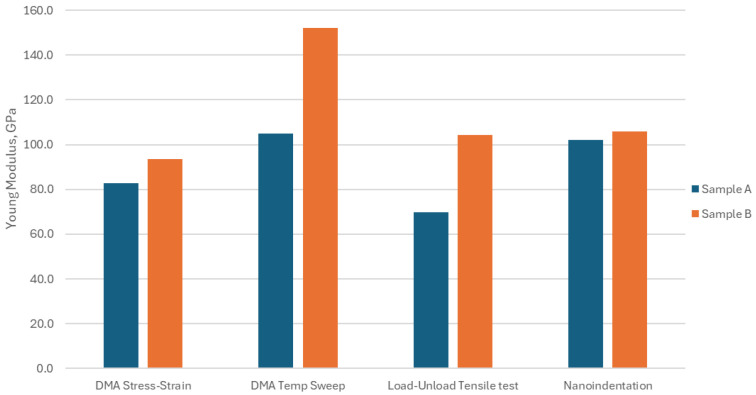
Modulus comparison using 4 test methods at 30 °C.

**Figure 8 micromachines-16-00428-f008:**
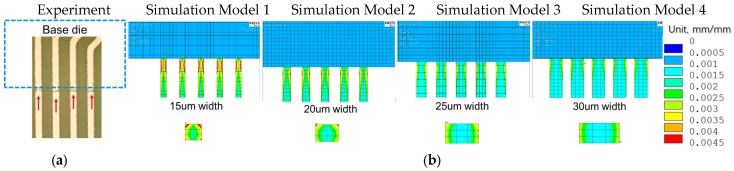
Effect of Cu trace width on reliability. (**a**) Cu trace crack highlighted by red arrows near base die edge after TC-C 1000 cycles for 20 µm Cu trace width. (**b**) Simulation results of plastic strain with different copper trace widths, showing plastic strain distribution of Cu traces located at base die edge in the top view and cross-sectional view.

**Figure 9 micromachines-16-00428-f009:**
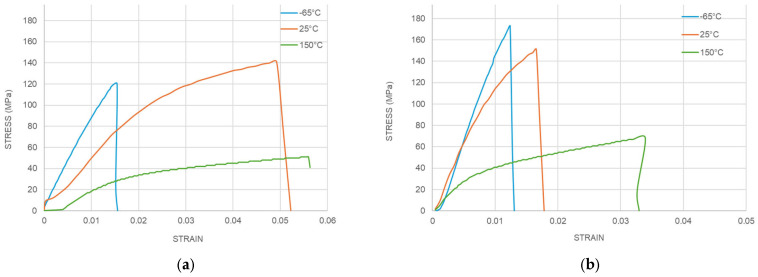
Stress–strain behavior of SR at different temperatures. (**a**) SR type 1. (**b**) SR type 2.

**Figure 10 micromachines-16-00428-f010:**
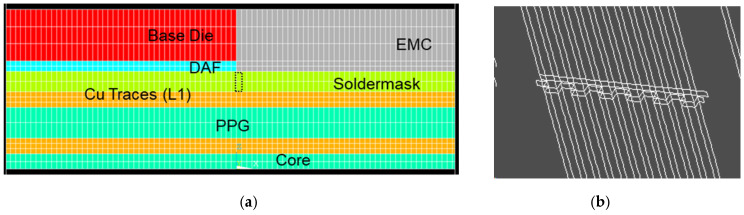
Progressive model using killed SR elements near base die edge: (**a**) side view and (**b**) isotropic view.

**Figure 11 micromachines-16-00428-f011:**
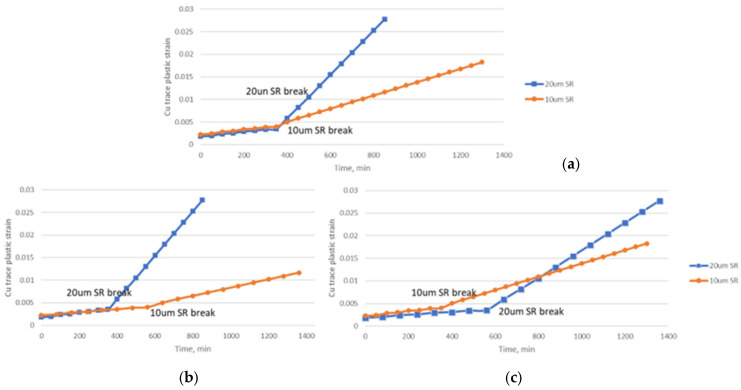
Cu trace plastic strain accumulation before and after SR breaks, assuming (**a**) 10 µm and 20 µm breaks at the same time, (**b**) 20 µm SR breaks first, and (**c**) 10 µm SR breaks first.

**Table 1 micromachines-16-00428-t001:** Substrate copper trace crack failure modes under TCTs.

Failure Modes	A	B1	B2	C
Schematic	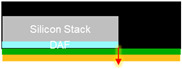	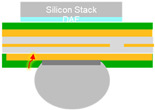	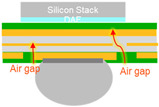	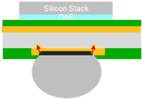
SEM Images	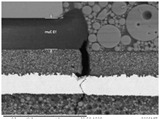	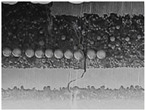	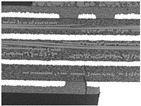	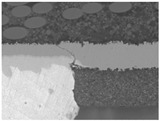
Initiation	Top solder resist (SR)	Bottom SR	Bottom SR	IMC/Cu interface
Propagation	SR ⟶ Cu	SR ⟶ Cu	SR ⟶ PPG ⟶ Cu	IMC ⟶ Cu
Location (Z)	Primary	Secondary layer	Inner layer or primary layer	Secondary layer
Location (XY)	Base die edge	Die edge/shadow	Die edge/shadow	Die shadow

**Table 2 micromachines-16-00428-t002:** Copper foil sample information.

	Sample A	Sample B
Supplier	A	B
ED copper foil thickness, µm	1.5	2
Electrolytic-plated copper thickness, µm	13.5	13
Total copper sample thickness, µm	15	15
Annealing temperature	190 °C	150 °C

**Table 3 micromachines-16-00428-t003:** Copper foil characterization methods and conditions.

	Test Method	Sample Dimension	Test Temperature	Test Condition	Sample Size
1	Tensile Stress–Strain	3 × 10 mm	−65 °C, 30 °C, 150 °C	0.02%/s	3
2	Tensile Temperature Sweep	3 × 10 mm	−65 °C~260 °C	3 °C/min, 1 Hz	3
3	Tensile Loading–Unloading	3 × 50 mm	−65 °C, 30 °C, 150 °C	0.2%/s	3
4	Nanoindentation	-	30 °C	300 nm depth	16

**Table 4 micromachines-16-00428-t004:** Young’s modulus from stress–strain curves obtained using DMA tensile tests.

Young’s Modulus, GPa	−65 °C	30 °C	150 °C
Sample A	90.0	82.6	60.2
Sample B	106.2	93.5	90.8

**Table 5 micromachines-16-00428-t005:** Young’s modulus from the DMA temperature sweep test.

Storage Modulus, GPa	−65 °C	30 °C	150 °C
Sample A	111	105	89
Sample B	161	152	132

**Table 6 micromachines-16-00428-t006:** Young’s modulus by loading–unloading test.

Young’s Modulus, GPa	−65 °C	30 °C	150 °C
Sample A	63	70	66
Sample B	87	104	96

**Table 7 micromachines-16-00428-t007:** Young’s modulus and hardness by nanoindentation.

	Young’s Modulus (E), GPa	Hardness (H), GPa	E/H
Sample A	101 ± 3.5	1.37 ± 0.09	74
Sample B	106 ± 2.9	1.73 ± 0.08	61

**Table 8 micromachines-16-00428-t008:** Simulation correlation with experiment in terms of failure location.

Failure Mode	A	B	C
Experimental Results	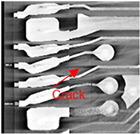	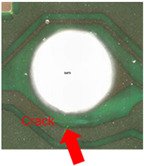	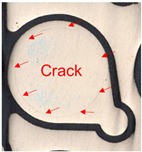
Simulation Results	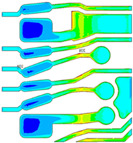	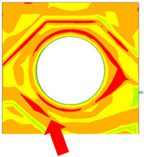	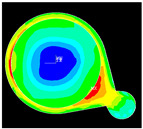

**Table 9 micromachines-16-00428-t009:** Simulation results for different copper trace widths.

Model	1	2	3	4
Trace Width, µm	15	20	25	30
Trace Thickness, µm	15	15	15	15
Relative Cu Plastic Strain	127%	109%	100%	87%

**Table 10 micromachines-16-00428-t010:** Simulation results for different copper dimensions and inputs of material properties.

	Model 4	Model 2a	Model 2b
	Supplier A: actual case		Supplier B: actual case
Measured trace width, µm	30	20	20
Measured trace thickness, µm	15	13	13
Cross-sectional area, µm^2^	450	260	260
Stress–strain input	Supplier A	Supplier A	Supplier B
Relative Vol. Ave. plastic strain (simulation)	100%	122%	152%
TC-C 1000 cycles results (experiment)	Pass	-	Fail

**Table 11 micromachines-16-00428-t011:** Effect of SR type from simulation and experimental perspectives.

			Design A with SR Type 1	Design B with SR Type 2
Simulation Response	Failure mode A	Stress on the top SR	100%	66%
		Strain on the top Cu trace	100%	25%
	Failure mode B2	Stress on the bottom SR	100%	67%
		Strain on the inner Cu trace	100%	97%
Experiment Failure Rate	Failure mode A	TC-C 2000 cycles	80/240	25/240
	Failure mode B2	TC-C 1000 cycles	20/79	0/79

## Data Availability

The original contributions presented in this study are included in the article. Further inquiries can be directed to the corresponding authors.

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
