# Peer review of "An Overview of Substrate Copper Trace Crack Through Experiments, Characterization, and Numerical Simulations"

_micromachines, 2025, doi:10.3390/mi16040428_

Round 1
Reviewer 1 Report
Comments and Suggestions for Authors
Dear Authors,
please find my review of the manuscript: "An Overview of Substrate Copper Trace Crack Through Experiment, Characterization and Numerical Simulation" below.
1. I suggest to add some quantitative data to the abstract, so that the findings are more grounded for the reader even in the abstract. The same goes to conclusions. Just an example: "There is strong correlation" - if so, please characterize numerically.
2. I suggest a harmonization between FIG1-2 and FIG3 display methods.
3. It is not clear if Samples A and B are two different (let's say) sheets, which are prepared for a larger amount of smaller samples, or how they are handled. I see sample dimensions in Table 3, but there is a need to show sample count, and maybe related standards (or citations) on how the count relates to the usually acknowledged result. This should be investigated up to figure 7...
4. Simulation response images misses legend highlights.
5. Simulation response images have more "hot spots" than where actual failure happens. Please discuss.
6. There is a need to discuss literature findings with current goals - this aspect need further elaboration.
7. The English and the overall presentation of the paper is acceptable.
Author Response
Response to Reviewer 1:
Reviewer #1
- I suggest to add some quantitative data to the abstract, so that the findings are more grounded for the reader even in the abstract. The same goes to conclusions. Just an example: "There is strong correlation" - if so, please characterize numerically.
Answer: Thanks for your comments, we added quantitative data to the abstract and conclusion, please refer to the revision. Here shows some examples:
- Cu trace width changing from 15 µm to 30 µm results in 40% drop for plastic strain range on Cu trace.
- Simulation results on different material properties show 30% difference in plastic strain.
- Effect of SR type can reduce SR stress and Cu plastic strain up to 75%.
- Plastic strain with and without solder resist protection has four times difference.
- I suggest a harmonization between FIG1-2 and FIG3 display methods.
Answer: We amened Figure 3 with similar format of Figure 2 using image quality map and grain size image.
- It is not clear if Samples A and B are two different (let's say) sheets, which are prepared for a larger amount of smaller samples, or how they are handled. I see sample dimensions in Table 3, but there is a need to show sample count, and maybe related standards (or citations) on how the count relates to the usually acknowledged result. This should be investigated up to figure 7...
Answer: Thanks for your comments. Yes, sample A and sample B are 2 different sheets, which have different process setting and annealing treatment from 2 different substrate vendors. We carefully labeled and handled samples for different types of tests and results analysis. Sample size for each test follows recommendation by ASTM standards, such as 3 samples for tensile test to obtain averaged results to reduce result variation. We added sample size in table 3 and highlighted averaged results for tensile tests and added error for nanoindentation due to more data available (16 testing points for each sample).
Tensile test including DMA stress-strain, DMA temperature sweep, Loading-unloading follows test standard: ASTM E345-16.
Figure 7 is removed because it is a normal testing curve by nanoindentation CSM testing method and we added +/- range for test results of modulus and hardness in Table 7 to reflect consistent and good testing results over 16 tested points.
- Simulation response images misses legend highlights.
Answer: Thanks for your comments. In Table 8, the purpose of simulation results is to correlate high stress/strain (red color in result contour) location with failure location for 3 different failure modes. It is just relative comparison between simulation results and failure. So, the legend is not used in Table 8.
We added legend for strain value in Figure. 8 on the study of Cu trace width, please refer to the revision for the detail.
- Simulation response images have more "hot spots" than where actual failure happens. Please discuss.
Answer: in Table 8 correlation between simulation and experimental failure location: simulation shows multiple “hot spots” (red color showing high stress/strain location from simulation), which are potential failure location. However, in actual failure analysis, failure site may only one site such as mode B. The reason is that when the crack happens, energy releases and stress concentrates in the crack tips and stress in other location reduces. Simulation results on stress is static analysis results without considering dynamic situation in the real experiment. The potential cracking location has been correlated well by simulation results for such scenario (mode B). For mode C, cracks can initiate at both high stress location and propagate to large cracks and connect. Therefore, the static simulation stress with hot spot (high stress location) can be explained as potential crack initiation location, which is the focus in this study.
- There is a need to discuss literature findings with current goals - this aspect need further elaboration.
Answer: Thanks for your comments. In the revision we added more discussion on literature finding with our results. We also added more references. Please refer to revision for more information.
- The English and the overall presentation of the paper is acceptable.
Answer: Thanks for your valuable comments!

Reviewer 2 Report
Comments and Suggestions for Authors
The primary concern of this paper is the fatigue reliability of Cu traces in PCB assessed by FEM simulation. To enable the computational analysis of the fatigue, it also conducted experimental measurements on the properties related to the fatigue performance of the trace, namely Young's modulus, yield strength, and hardness of the chosen samples. The approaches presented in this paper is reasonable and consistent with many previous studies, while adding no insights to the subject. The interesting part is the finding of SR impacting the fatigue performance of Cu traces. In this respect, this paper can be of interest to many but is not without issues including but not limited to:
1) editorial: this paper is not easy to follow not because of English but because of not so great flow of ideas. The introduction reviews a lot of Cu as interconnect but not much discussion about its properties when used as traces in PCB, fatigue mechanisms, and what this paper addresses. The scope of paper becomes clear in about half way into the result section. It may be necessary to restructure the paper so that it is easier to follow. Also necessary is to conduct careful review of statements to make them carry clear meaning. This is especially needed in the introduction. Statement like "show how copper ion concentration impacts microstructure and mechanical properties" may not be easy to understand to non-expert. It is certain that the statement refers to the chemistry of electroplating and/or electroless plating, but without explicit statement it becomes a source of confusion. There are a few places of such problems. It is also necessary to state that this paper is about Cu in PCB as early as possible to quickly guide attention of readers to the main subject.
2. Technical; one of the main distractions is the differing Young's modulus for A and B. Young's modulus is an intrinsic property of materials and is not sensitive function of external factors like grain structure and solute in the Cu matrix. The result of sample B showing consistently higher Young's modulus than sample A is very puzzling and cannot be related to the Cu (because Cu is presumably poor to a level to have the same modulus). What is suspected is the core materials. Resin and fiberglass may be with different structure or arrangement to impact the modulus of Cu traces when experimentally measured. Since the whole FEA analysis is based on modulus and yield strength experimentally measured, any result leading to the conclusion of this paper may not free from the error. Careful review of the data and discussion of possible reason of difference among A and B is highly recommended.
Author Response
Response to Reviewer 2:
Reviewer #2
The primary concern of this paper is the fatigue reliability of Cu traces in PCB assessed by FEM simulation. To enable the computational analysis of the fatigue, it also conducted experimental measurements on the properties related to the fatigue performance of the trace, namely Young's modulus, yield strength, and hardness of the chosen samples. The approaches presented in this paper is reasonable and consistent with many previous studies, while adding no insights to the subject. The interesting part is the finding of SR impacting the fatigue performance of Cu traces. In this respect, this paper can be of interest to many but is not without issues including but not limited to:
Answer: Thanks for your valuable comments! We agree that Cu foil material properties are studied by many researchers previously. Our research provides some insight into Cu trace fatigue failure mechanisms such as grain size and microstructure effect, understanding on SR protection on Cu trace. Accurate multi-linear elastic-plastic properties used in FEA simulation are obtained using Cu foil sample directly from substrate vendor with the same process setting as their laminated substrate manufacturing and then stress-strain response from simulation is correlated well with experimental results in terms of failure location. We revised manuscript accordingly by considering your comments.
Please refer to revision for more details.
- editorial: this paper is not easy to follow not because of English but because of not so great flow of ideas. The introduction reviews a lot of Cu as interconnect but not much discussion about its properties when used as traces in PCB, fatigue mechanisms, and what this paper addresses. The scope of paper becomes clear in about half way into the result section. It may be necessary to restructure the paper so that it is easier to follow. Also necessary is to conduct careful review of statements to make them carry clear meaning. This is especially needed in the introduction. Statement like "show how copper ion concentration impacts microstructure and mechanical properties" may not be easy to understand to non-expert. It is certain that the statement refers to the chemistry of electroplating and/or electroless plating, but without explicit statement it becomes a source of confusion. There are a few places of such problems. It is also necessary to state that this paper is about Cu in PCB as early as possible to quickly guide attention of readers to the main subject.
Answer: Thanks for your valuable comments. We rewrite introduction with more explicit statement for references and highlight objectives of the study. Please refer to the revision for more details.
Here shows an example of purpose of this paper and state them in first paragraph for quick attention of readers:
The study aims to provide an overview of the package substrate copper trace crack mechanism and prevention through package level reliability experiments, Cu foil material characterization, and numerical simulations on package reliability assessment subjected to TCT.
- Technical; one of the main distractions is the differing Young's modulus for A and B. Young's modulus is an intrinsic property of materials and is not sensitive function of external factors like grain structure and solute in the Cu matrix. The result of sample B showing consistently higher Young's modulus than sample A is very puzzling and cannot be related to the Cu (because Cu is presumably poor to a level to have the same modulus). What is suspected is the core materials. Resin and fiberglass may be with different structure or arrangement to impact the modulus of Cu traces when experimentally measured. Since the whole FEA analysis is based on modulus and yield strength experimentally measured, any result leading to the conclusion of this paper may not free from the error. Careful review of the data and discussion of possible reason of difference among A and B is highly recommended.
Answer: Thanks for your comments. We agree that Young's modulus is an intrinsic property of materials. However, its value is affected by manufacturing process, which causes different microstructures such as grain size and orientation. These microstructure effect on material properties have been investigated and validated by many researchers, such as study in reference 1 (we discussed in introduction and cited it)
Researchers have explored intrinsic properties of copper foil, with process parameters significantly affecting its microstructure, which in turn influences properties like coefficient of thermal expansion (CTE), mechanical properties, and fatigue behavior [1].
In this study, pure Cu foil samples were used for material characterization, as shown in the last step of copper foil sample in the revised figure 1. There is no core, resin and fiberglass in tested sample. Sorry for confusion based on original figure 1. We provide schematic of process flow for sample preparation of copper foil in updated figure 1.
Samples A and B are prepared by 2 different substrate vendors with their own process parameter setting and geometry and annealing condition are shown in Table 2. Other process parameters are not listed due to confidentiality. However, microstructure differences (grain size) as shown in Figure 5 investigated by us results in different properties such as modulus, yield and hardness. The results are consistent with fundamental and other researchers’ findings. FEA analysis using tested Cu properties provided consistent results with experimental data of real packages: supplier B has poorer reliability with Cu trace cracking in package level TCT test as compared to supplier A due to higher Cu plastic deformation.
In the revision, we added more discussion on difference between sample/supplier A and B.
Your sincerely,
Wei Yu, Faxing Che, etc.

Round 2
Reviewer 1 Report
Comments and Suggestions for Authors
Dear Authors, i have a few minor remarks.
0. Most responses are convincing, thank you for the thorough work.
- The simulation hot spot issue is also good, however i miss some statistical mindset on this. If i test this issue on X boards, will the boards fail always on the same spot? Is there a correlation between absolute "red" values, vs crack location? Please give a short discussion. Also this raises a second question, there is no legend in table 8 images, and figure 8 legend is non-readable without the legends units. Please export vector form images for figure 8.
- Overall the rest of the paper is acceptable now.
Reviewer 2 Report
Comments and Suggestions for Authors
The revised reads better.